# Efficacy of Liver-Directed Combined Radiotherapy in Locally Advanced Hepatocellular Carcinoma with Portal Vein Tumor Thrombosis

**DOI:** 10.3390/cancers15123164

**Published:** 2023-06-13

**Authors:** Jina Kim, Jason Chia-Hsien Cheng, Taek-Keun Nam, Jin Hee Kim, Byoung Kuk Jang, Wen-Yen Huang, Hiroshi Aikata, Myungsoo Kim, Jung Hyun Kwon, Jinbo Yue, Victor Ho Fun Lee, Zhaochong Zeng, Jinsil Seong

**Affiliations:** 1Department of Radiation Oncology, Yonsei Cancer Center, Yonsei University College of Medicine, 50-1 Yonsei-ro, Seodaemun-gu, Seoul 03722, Republic of Korea; jinabelle@yuhs.ac; 2Department of Radiation Oncology, National Taiwan University Hospital, Taipei 100229, Taiwan; jasoncheng@ntu.edu.tw; 3Department of Radiation Oncology, Chonnam National University Hwasun Hospital, Hwasun 58128, Republic of Korea; tknam@chonnam.ac.kr; 4Department of Radiation Oncology, Keimyung University, Dongsan Hospital, Daegu 42601, Republic of Korea; jhkim@dsmc.or.kr; 5Department of Internal Medicine, Keimyung University, Dongsan Hospital, Daegu 42601, Republic of Korea; jangha106@dsmc.or.kr; 6Department of Radiation Oncology, Tri-Service General Hospital, National Defense Medical Center, Taipei 114202, Taiwan; hwyyi@yahoo.com.tw; 7Department of Gastroenterology, Hiroshima Prefectural Hospital, Hiroshima 734-0004, Japan; aikatahiroshi@icloud.com; 8Department of Radiation Oncology, Incheon St. Mary’s Hospital, College of Medicine, The Catholic University of Korea, Seoul 21431, Republic of Korea; mskim0710@catholic.ac.kr; 9Division of Hepatology, Department of Internal Medicine, Incheon St. Mary’s Hospital, College of Medicine, The Catholic University of Korea, Seoul 21431, Republic of Korea; doctorkwon@catholic.ac.kr; 10Department of Radiation Oncology, Shandong Cancer Hospital and Institute, Shandong First Medical University and Shandong Academy of Medical Sciences, Jinan 250117, China; jbyue@sdfmu.edu.cn; 11Department of Clinical Oncology, The University of Hong Kong, Hong Kong SAR, China; vhflee@hku.hk; 12Department of Radiation Oncology, Zhongshan Hospital Affiliated to Fudan University, Shanghai 200031, China; zeng.zhaochong@zs-hospital.sh.cn

**Keywords:** hepatocellular carcinoma, portal vein tumor thrombosis, sorafenib, radiotherapy, prognostic factors

## Abstract

**Simple Summary:**

In this multinational, multi-institutional study, we investigated the efficacy of liver-directed combined radiotherapy compared with sorafenib in hepatocellular carcinoma patients presenting portal vein tumor thrombosis. Propensity score matching was performed to minimize the imbalance between the two groups. The median overall survival was significantly improved in the LD combined RT group, and the conversion rate to curative surgery was also significantly higher in the LD combined RT group. Despite the multimodality of the treatments, toxicity rates of LD combined RT were comparable to those of sorafenib.

**Abstract:**

Purpose: Although systemic treatment is the mainstay for advanced hepatocellular carcinoma (HCC), numerous studies have highlighted the added value of local treatment. This study aimed to investigate the clinical efficacy of liver-directed combined radiotherapy (LD combined RT) compared with that of sorafenib, a recommended treatment until recently for locally advanced HCC presenting portal vein tumor thrombosis (PVTT), using a multinational patient cohort. Materials and Methods: We identified patients with HCC presenting PVTT treated with either sorafenib or LD combined RT in 10 tertiary hospitals in Asia from 2005 to 2014. Propensity score matching (PSM) was performed to minimize the imbalance between the two groups. The primary endpoint was overall survival (OS), and the secondary endpoints were progression-free survival (PFS) and treatment-related toxicity. Results: A total of 1035 patients (675 in the LD combined RT group and 360 in the sorafenib group) were included in this study. After PSM, 305 patients from each group were included in the analysis. At a median follow-up of 22.5 months, the median OS was 10.6 and 4.2 months for the LD combined RT and sorafenib groups, respectively (*p* < 0.001). The conversion rate to curative surgery was significantly higher (8.5% vs. 1.0%, *p* < 0.001), while grade ≥ 3 toxicity was fewer (9.2% vs. 16.1%, *p* < 0.001) in the LD combined RT group. Conclusions: LD combined RT improved survival outcomes with a higher conversion rate to curative surgery in patients with locally advanced HCC presenting PVTT. Although further prospective studies are warranted, active multimodal local treatment involving radiotherapy is suggested for locally advanced HCC presenting PVTT.

## 1. Introduction

Hepatocellular carcinoma (HCC), one of the most common malignant tumors, ranks third as the leading cause of cancer-related deaths worldwide and poses a great challenge to both patients and physicians [1,2]. Portal vein tumor thrombosis (PVTT) is a well-known poor prognostic factor for patients with HCC, with a reported median survival of 2.7–4.0 months if left untreated [3,4]. While the incidence is quite high, ranging from 10% to 60% [5,6], current guidelines only recommend the use of systemic treatment for this advanced stage [7]. However, the degree of PVTT can be heterogeneous (from focal to spread to the main trunk), which is related to a wide range of prognoses [8].

While the treatment of HCC with PVTT remains focused on systemic treatment, the value of adding local treatment has been suggested by numerous studies. Transcatheter arterial chemoembolization (TACE) showed efficacy compared with that of conservative treatment in all PVTT types [9]. Surgical resection has proven beneficial in selected patients—usually those with a minimal extent of PVTT [10]. Liver-directed radiotherapy (RT) has been shown to be an effective treatment option for HCC in numerous reports [1,11,12,13]. RT with or without TACE showed improved survival to sorafenib in Barcelona clinic liver cancer stage C disease [14,15], and RT combined with TACE was superior to surgical resection by a median survival of 2.3 months [16]. Concurrent chemoradiotherapy (CCRT) combined with hepatic arterial infusion chemotherapy (HAIC) has also proved to be superior to sorafenib in patients with HCC presenting PVTT [17].

Numerous local treatment strategies, including TACE, transarterial radioembolization, percutaneous ethanol injection, and radiofrequency ablation (RFA), have been developed over the years, all which greatly improved the survival outcomes of HCC patients, and they offer various options for patients and physicians [18,19,20]. Attempts at aggressive local treatment for HCC with PVTT have been made over the years, but comparative studies with sorafenib, a recommended treatment until recently, and multimodal local treatments are limited. Therefore, this study was designed to compare the clinical outcomes of sorafenib and liver-directed combined RT (LD combined RT) in patients with HCC presenting PVTT. This multi-institutional study included patients treated with various modalities, including sorafenib, TACE plus RT, RFA plus RT, and CCRT, and presented the results of a comprehensive analysis.

## 2. Materials and Methods

### 2.1. Study Population

The medical records of patients with HCC treated between January 2005 and November 2014 at 10 hospitals (4 in Korea, 2 in Taiwan, 2 in China, 1 in Hong Kong, and 1 in Japan) were reviewed for this study. The patient inclusion criteria were as follows: (a) age ≥ 20 years at the time of diagnosis, (b) histologically or radiologically confirmed HCC with PVTT, (c) disease confined within the liver, and (d) treated with sorafenib or LD combined RT. Patients who received sorafenib with RT, those with an unknown PVTT extent, those lost to follow-up, or those with tumor thrombosis extending to the heart were excluded. A total of 1035 patients (675 of whom received LD combined RT and 360 of whom received sorafenib treatment) were included (Figure 1). This study was approved by the institutional review boards of all participating institutions (approval no. 4-2020-0498), and the requirement for informed consent was waived because of the retrospective study design.

### 2.2. Treatment

Treatment was decided on an institutional basis, considering each patient’s age, performance status, liver function, prior treatment history, and disease extent. Patients in the sorafenib group initially received 400 mg sorafenib twice a day, and the dose was modified to 400 mg sorafenib once a day or 200 mg once a day if adverse reactions were noted. For patients in the LD combined RT group, RT was administered either alone or in combination with other local (such as TACE, RFA, or HAIC) therapies. When in combination, LD combined RT targets the main lesion including PVTT, whereas TACE or RFA target satellite lesions. Regarding the RT modalities, 552 patients (81.8%) received 3D-conformal RT, 113 (16.7%) received intensity-modulated RT, and 10 (1.5%) received stereotactic body RT. The median RT dose was 45 Gy (interquartile range [IQR] 45–50 Gy) with 1.8–3 Gy per fraction, which translates to a median biological effective dose of 53.1 Gy (IQR, 53.1–60 Gy). For patients who underwent CCRT, concurrent HAIC with 5-fluorouracil (500 mg/m^2^/day) was administered in the first and last weeks of the 5-week RT course. The selected patients underwent surgery of either liver resection or transplantation upon tumor regression after initial treatment.

### 2.3. Response Evaluation and Statistical Analysis

For tumor response assessment, follow-up imaging studies were performed every 1–3 months and judged based on the modified Response Evaluation Criteria in Solid Tumors (mRECIST). Treatment-related toxicity was evaluated based on the Common Terminology Criteria for Adverse Events (CTCAE) version 4.0.

The primary endpoint was overall survival (OS), and the secondary endpoints were progression-free survival (PFS) and treatment-related toxicity. We used two different time points for survival analyses: one was defined as the time from treatment initiation and the other as the time from diagnosis. The time from treatment initiation was used primarily, unless otherwise specified. The Kaplan–Meier method was used for OS and PFS analyses, and the Cox proportional hazard regression model was used for univariable and multivariable analyses.

The baseline characteristics of the two groups were compared using the chi-square test, Fisher’s exact test, or Student’s *t*-test. We used propensity score matching (PSM) to overcome differences in patient and tumor characteristics. A 1:1 PSM with the nearest-neighbor method and a caliper width of 0.2 was conducted to account for statistical differences in sex, age, Eastern Cooperative Oncology Group (ECOG) performance status, prior treatment history, tumor size, disease extent, and PVTT extent between the two groups. For the description of PVTT extent, Cheng’s classification was applied in this study: (a) type I, tumor thrombosis in the segmental or sectoral branches of the portal vein; (b) type II, tumor thrombosis in the right and/or left portal vein; (c) type III, tumor thrombosis involving the main portal vein; and (d) type IV, tumor thrombosis beyond the main portal vein involving the superior mesenteric vein [21]. Statistical significance was set at *p* < 0.05. SPSS Statistics for Windows (version 26.0; IBM Corp., Armonk, NY, USA) was used for all the statistical analyses.

## 3. Results

The patient and tumor characteristics before and after PSM are shown in Table 1. Of the entire cohort, 898 patients (86.8%) were male, and the median age was 57 (IQR 50–65). Approximately three-quarters of the patients (772 patients, 74.5%) had a well-preserved liver function of Child–Pugh class A, and most patients had a good performance status of ECOG ≤ 1 (942 patients, 91.0%). Compared with that of the LD combined RT group, the sorafenib group had more females (16.4% vs. 11.6%, *p* = 0.029), a higher median age (60 vs. 55, *p* < 0.001), more patients with an ECOG performance score of 2 or higher (15.0% vs. 5.7%, *p* < 0.001), more patients with prior treatment history (47.2% vs. 20.4%, *p* < 0.001), a smaller tumor size (8.2 cm vs. 9.6 cm, *p* < 0.001), more bilateral disease (61.1% vs. 36.7%, *p* < 0.001), and more patients with PVTT confined to the segmental branches of the portal vein (6.7% vs. 2.2%, *p* = 0.002). After PSM, 305 patients from each group were matched, and no differences in the aforementioned characteristics were observed. Regarding the details of LD combined RT, CCRT was the most common (502 patients, 74.4%), followed by TACE plus RT (149 patients, 22.1%) and TACE plus RFA plus RT (18 patients, 2.7%) prior to PSM. After PSM, 201 (65.9%), 88 (28.9%), and 13 (4.3%) patients received CCRT, TACE plus RT, and TACE plus RFA plus RT, respectively.

At a median follow-up of 22.5 months (IQR, 11.3–44.5 months), the median OS as defined from treatment initiation was 10.6 and 4.2 and months for the LD combined RT and sorafenib groups, respectively (*p* < 0.001) (Figure 2A). The LD combined RT group also showed superior survival outcomes in terms of PFS: the median PFS as defined from treatment initiation was 8.1 months in the LD combined RT group and 3.1 months in the sorafenib group (*p* < 0.001) (Figure 2B). The LD combined RT group also showed superior OS and PFS compared to those of the sorafenib group, as defined from diagnosis (Figure 2C,D).

The conversion rate to curative surgery of either liver resection or transplantation after the initial treatment was significantly higher in the LD combined RT group [26 patients (8.5%) vs. 3 patients (1.0%), *p* < 0.001]. Patients who underwent surgery had a younger median age (55 vs. 58 years, *p* = 0.048), smaller median tumor size (7.0 cm vs. 8.5 cm, *p* = 0.002), and more unilateral disease (75.9% vs. 45.1%, *p* = 0.001) than that of those who did not undergo surgery. Patients who underwent surgery of either type had significantly improved OS and PFS (Appendix A). The median OS was 48.7 months for those who underwent surgery and 7.0 months for those who did not (*p* < 0.001).

Subgroup analysis revealed that LD combined RT was beneficial for both patients with or without prior treatment history in terms of OS (Figure 3A,B). In patients with no prior treatment history, the median OS was 11.4 months and 3.6 months for those who received LD combined RT and sorafenib, respectively (*p* < 0.001). For patients with a prior treatment history, the median OS was 9.3 months and 5.0 months for those who received LD combined RT and sorafenib, respectively (*p* = 0.037). When subgroup analysis was performed according to PVTT type, patients with PVTT types II and III showed significantly improved OS after LD combined RT (Appendix A). In addition, patients benefited from LD combined RT regardless of alpha-fetoprotein (AFP) level (≤400 ng/mL or >400 ng/mL) in terms of OS (Appendix A).

Univariable analysis revealed that treatment type, ECOG performance status, Child–Pugh class, pretreatment AFP level, tumor size, disease extent, lymph node metastasis, and PVTT type were prognostic factors for OS both before and after PSM. Among these factors, treatment type, Child–Pugh class, pretreatment AFP level, and tumor size were statistically significant in the multivariable analysis (Table 2). Regarding PFS, treatment type, ECOG performance status, Child–Pugh class, pretreatment AFP level, tumor size, disease extent, and lymph node metastasis were prognostic factors in the univariable analysis (Appendix A). Among these factors, treatment type, Child–Pugh class, pretreatment AFP level, and tumor size remained significant in the multivariable analysis.

For patients in the LD combined RT group, aspartate aminotransferase (AST)/alanine aminotransferase (ALT) elevation (10.8%) was the most common treatment-related acute toxicity, followed by bilirubin elevation (9.5%), abdominal pain (5.8%), and anorexia (4.0%). For patients in the sorafenib group, the most common treatment-related acute toxicity was diarrhea (18.1%), followed by AST/ALT elevation (13.0%), bilirubin elevation (11.7%), skin rash (10.8%), and hand–foot syndrome (7.8%) (Table 3). Grade three or higher toxicity was noted less frequently in the LD combined RT group than that in the sorafenib group (9.2% vs. 16.1%, *p* < 0.001).

## 4. Discussion

In this multinational study, we investigated the clinical efficacy of LD combined RT compared with that of sorafenib in 1035 patients with liver-confined HCC presenting PVTT. The LD combined RT group showed significantly superior outcomes in terms of OS and PFS and over an eight-fold higher conversion rate to curative surgery than that in the sorafenib group. LD combined RT also showed favorable treatment-related toxicity profiles, with both grade three or higher acute and late toxicities occurring more frequently in the sorafenib group.

In our study, the median OSs were 10.6 and 4.2 months in the LD combined RT and sorafenib groups, respectively (*p* < 0.001) (Figure 2A). A previous study analyzing the effectiveness of HAIC combined with RT in advanced HCC patients presenting PVTT observed a median OS of 12.1 months, which is 1.5 months longer compared to that of the LD combined RT group in this study [22]. Moreover, a recently conducted IMbrave150 study, which compared the efficacy of atezolizumab plus bevacizumab with that of sorafenib in patients with unresectable HCC, reported a median OS of 13.2 months in the sorafenib group, which was much longer than that of the sorafenib group in this study [23]. While our data showed a shorter median OS compared to the two studies, it is important to note that the two studies only included patients with an ECOG performance status of zero or one, Child–Pugh class of A to B7, and those with no prior history of systemic treatment, whereas our study also included patients with an ECOG performance status of two or three, Child–Pugh class of B8 or B9, and patients with prior treatment history. In addition, only about 40% of patients in the IMbrave150 study had macrovascular invasion, while all patients in our study had PVTT.

Aggressive treatment of PVTT is necessary since PVTT leads to both intrahepatic and extrahepatic disease spread, portal hypertension and ascites, and liver function deterioration if left untreated [3,5]. A subgroup analysis of a phase III trial investigating the efficacy of sorafenib in patients with locally advanced HCC revealed that while sorafenib improved OS and time to progression compared with placebo, the benefit decreased in patients with macrovascular invasion [24]. On the other hand, a high response rate of PVTT to RT has been reported previously [12,25,26], and it could be further improved when multimodal treatment, such as HAIC in combination with RT, is administered [2]. The tumor downstaging rate of our group was comparable to that of other groups—a recent study from a Japanese group reported tumor downstaging in 11.8% of patients after combination therapy of HAIC and RT [22]. Adding RT to TACE showed significantly improved survival results to TACE alone in several meta-analysis reports [27,28,29]. One randomized clinical trial comparing sorafenib to TACE plus RT for patients with HCC with macroscopic vascular invasion revealed that the PFS and radiologic response rates were significantly higher in the TACE plus RT group [30]. A Japanese group investigated the efficacy of HAIC combined with RT for patients with HCC with tumor thrombosis in the main trunk or bilobar of the portal vein and reported a 51.0% response rate of the PVTT [22]. In another study comparing the outcomes of liver-directed CCRT to sorafenib, liver-directed CCRT showed superior OS (median OS 9.8 months vs. 4.3 months, *p* = 0.002) [17].

Univariable and multivariable analyses revealed that treatment type, Child–Pugh class, pretreatment AFP level, and tumor size were significantly predictive of prognosis. While other patient and tumor characteristics are non-modifiable at the time of diagnosis, treatment type is a modifiable factor that may improve prognosis. Subgroup analysis of our data showed that LD combined RT outperformed sorafenib with statistical significance in patients with PVTT types II and III but not in those with PVTT types I and IV (Appendix A). However, there were a limited number of patients with PVTT types I and IV included in this study, thus limiting the statistical power. On the other hand, both patients with or without a prior treatment history benefited from LD combined RT. Therefore, LD combined RT should be actively recommended for patients even with prior treatment history.

In our study, a much higher rate of patients in the LD combined RT group underwent surgery than those in the sorafenib group (8.5% vs. 1.0%, *p* < 0.001), and surgery after downstaging prolonged the survival outcomes. Similar to our study, a Chinese study of 116 patients reported that radical hepatectomy with thrombectomy prolonged the survival of patients with HCC presenting PVTT with low toxicity rates [31]. Another group also reported a statistically significant survival gain in patients with HCC presenting PVTT who received liver transplantations following RT compared with that of the RT alone group [11]. In a study comparing the outcomes of neoadjuvant RT followed by surgery and upfront surgery in patients with PVTT in the main portal vein, those who received neoadjuvant RT had significantly lower rates of HCC recurrence and HCC-related death [32]. This study underlines the importance of tumor downstaging prior to surgical resection. When patient characteristics were compared between those who underwent surgery and those who did not, patients who underwent surgery had a younger median age, smaller median tumor size, and more unilateral disease in our study. This is consistent with the findings of Lee et al., in which patients aged < 60 years, with a single tumor, no treatment history, pretreatment Child–Pugh class A, lower pretreatment tumor marker levels, and radiologic response after LD combined RT showed a higher conversion rate to surgery [33]. Thus, patients with such favorable features might be more often led to curative surgery by aggressive LD combined RT, resulting in improved survival outcomes. Given the non-negligible risk of RT-induced toxicity, we may only select patients with favorable features to undergo LD combined RT. Concerning that the improved survival outcomes of the LD combined RT group may have been due to the higher rate of patients who received surgery, we additionally performed a subgroup analysis of only patients who did not receive any surgery and observed significant OS improvement following LD combined RT compared to that of sorafenib (median OS 9.8 months vs. 4.2 months, *p* < 0.001, Appendix A).

Despite the aggressive multimodal treatment, the toxicity profile of the LD combined RT group was better than that of the sorafenib group. The sorafenib group reported higher rates of hand-foot syndrome, skin rash, diarrhea, and gastrointestinal bleeding than that of the LD combined RT group. Grade ≥ 3 toxicities were also more frequently reported in the sorafenib group in our study. One study from China compared the treatment outcomes of sorafenib plus RT and RT alone in 82 patients with HCC presenting PVTT and reported increased rates of grades 1–2 fatigue and skin reactions in the sorafenib plus RT group, with no RT-induced liver disease in either group [34]. Another study comparing the outcomes of TACE plus RT and TACE alone also reported comparable changes in liver function between the two groups [35]. Previous reports from our institution reported a 1.9% incidence of grade three nausea and stomach perforation and no grade four or higher toxicities following liver-directed CCRT [17]. These studies support that the toxicity rates of LD combined RT are comparable to those of sorafenib, despite the multimodality of treatments.

This study has some limitations, mainly due to its retrospective design. Patients in the LD combined RT group received various treatment combinations, giving rise to a heterogeneous study population. Few patients received RT alone or RFA plus RT, limiting the statistical analysis of the modality that would be the most effective. In addition, treatment-related toxicity may have been underreported. However, data from 10 institutions in 5 countries were collected and analyzed to minimize selection bias and strengthen statistical power. In addition, while atezolizumab plus bevacizumab has become the new norm for unresectable HCC following the results of the IMbrave150 study [23], the efficacy of LD combined RT was compared with that of sorafenib, a recommended treatment until recently, in this study. While this new systemic therapy regimen is more promising than previously used regimens, recent studies have suggested the clinical benefit of adding local treatment, such as RT, to this modern combination of systemic therapy [36].

## 5. Conclusions

In conclusion, our study suggests that LD combined RT improves survival outcomes in patients with locally advanced HCC presenting PVTT. While further prospective studies should be performed, we carefully suggest active multimodal local treatment, including RT, for locally advanced HCC presenting PVTT.

## Figures and Tables

**Figure 1 cancers-15-03164-f001:**
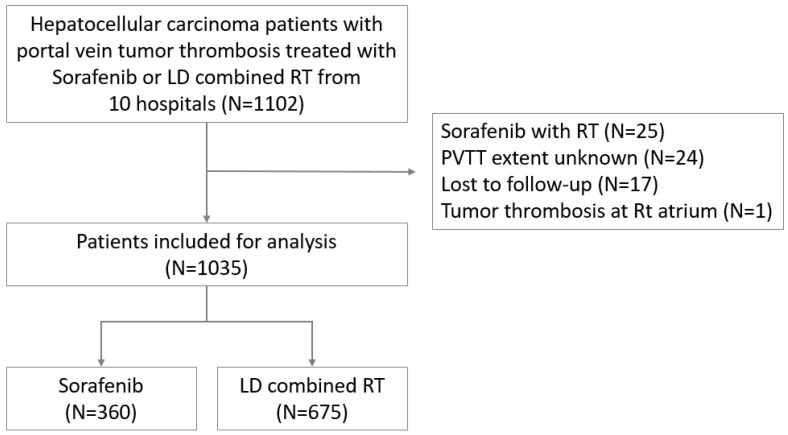
Study population.

**Figure 2 cancers-15-03164-f002:**
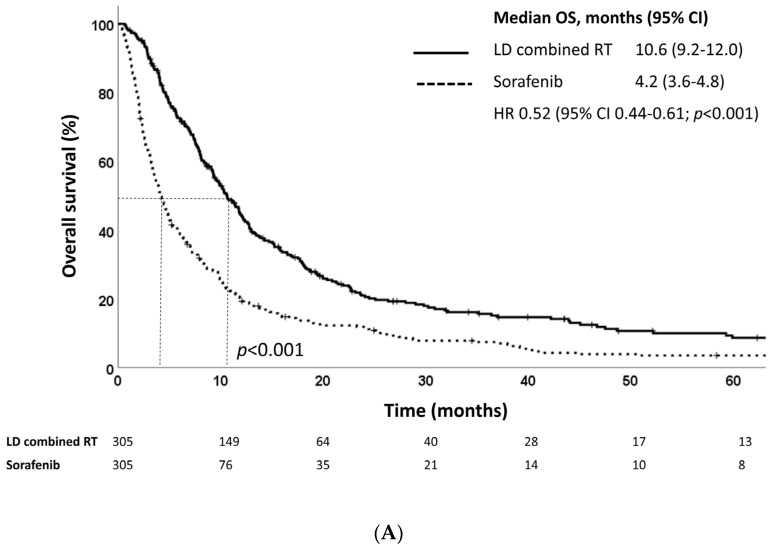
Kaplan–Meier estimates of (**A**) overall survival and (**B**) progression-free survival defined as time from treatment initiation, and (**C**) overall survival and (**D**) progression-free survival defined as time from diagnosis of LD combined RT and sorafenib groups in the propensity score-matched population.

**Figure 3 cancers-15-03164-f003:**
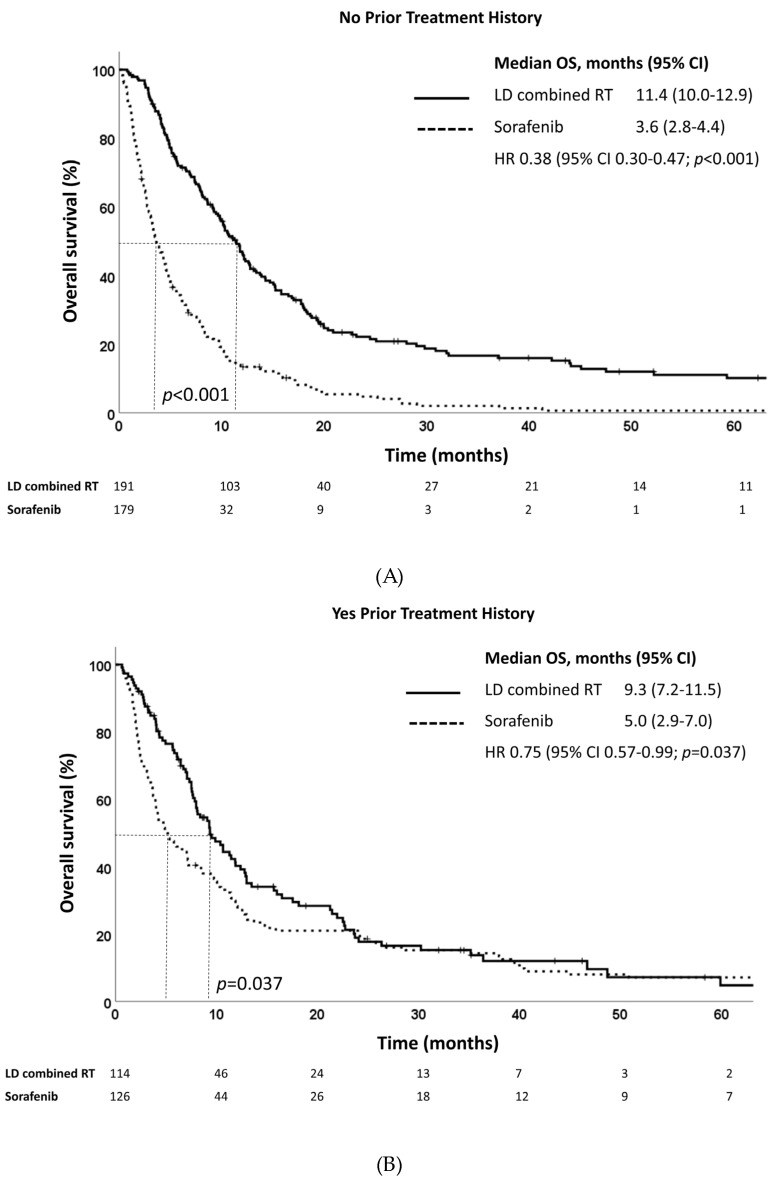
Overall survival estimates of LD combined RT and sorafenib in propensity score-matched patients with (**A**) no prior treatment history and (**B**) yes prior treatment history.

**Table 1 cancers-15-03164-t001:** Baseline characteristics of the sorafenib and liver-directed combined RT groups before and after propensity score matching.

	Before PSM			After PSM		
	Sorafenib	LD Combined RT		Sorafenib	LD Combined RT	
	(N = 360)	(N = 675)	*p* Value	(N = 305)	(N = 305)	*p* Value
Sex, n (%)			0.029			0.905
Female	59 (16.4)	78 (11.6)		41 (13.4)	40 (13.1)	
Median age [IQR], years	60 [52–68]	55 [49–63]	<0.001	59 [51–67]	57 [50–65]	0.090
ECOG, PS, n (%)			<0.001			0.344
0	86 (23.9)	179 (26.5)		82 (26.9)	67 (22.0)	
1	220 (61.1)	457 (67.7)		185 (60.7)	222 (72.8)	
2	40 (11.1)	37 (5.5)		30 (9.8)	15 (4.9)	
3	14 (3.9)	2 (0.2)		8 (2.6)	1 (0.3)	
Etiology, n (%)			0.174			0.536
HBV	252 (70.0)	522 (77.3)		218 (71.5)	220 (72.1)	
HCV	67 (18.6)	68 (10.1)		52 (17.0)	37 (12.1)	
NBNC	41 (11.4)	85 (12.6)		35 (11.5)	48 (15.7)	
Child–Pugh class, n (%)			0.123			0.777
A	268 (74.4)	504 (74.6)		230 (75.4)	233 (76.4)	
B	90 (25.0)	170 (25.2)		75 (24.6)	72 (23.6)	
C	2 (0.6)	1 (0.2)		0	0	
Prior treatment history, n (%)			<0.001			0.321
Yes	170 (47.2)	138 (20.4)		126 (41.3)	114 (37.4)	
Median AFP [IQR], ng/mL	1040.1 [42.6–17,218.8]	627.3 [31.0–11,572.2]	0.446	962.3 [45.7–15,769.0]	443.1 [26.7–9828.0]	0.602
Median tumor size [IQR], cm	8.2 [5.4–11.7]	9.6 [6.2–13.0]	<0.001	8.4 [5.5–11.9]	8.4 [5.3–11.0]	0.907
Disease extent			<0.001			0.257
Bilateral	220 (61.1)	248 (36.7)		170 (55.7)	156 (51.1)	
Lymph node status			0.848			0.257
Involved	47 (13.1)	91 (13.5)		40 (13.1)	31 (10.2)	
PVTT type (Cheng’s criteria)			0.002			0.945
I	24 (6.7)	15 (2.2)		16 (5.2)	12 (3.9)	
II	193 (53.6)	357 (52.9)		164 (53.8)	168 (55.1)	
III	138 (38.3)	297 (44.0)		121 (39.7)	124 (40.7)	
IV	5 (1.4)	6 (0.9)		4 (1.3)	1 (0.3)	

Abbreviations: PSM, propensity score matching; LD combined RT, liver-directed combined radiotherapy; IQR, interquartile range; ECOG PS, Eastern Cooperative Oncology Group performance status; HBV, hepatitis B virus; HCV, hepatitis C virus; NBNC, non-B, non-C; AFP, alpha-fetoprotein; PVTT, portal vein tumor thrombosis; CCRT, concurrent chemoradiotherapy; TACE, transcatheter arterial chemoembolization; RT, radiotherapy; and RFA, radiofrequency ablation.

**Table 2 cancers-15-03164-t002:** Prognostic factors for overall survival before and after PSM.

	Univariable Analysis			Multivariable Analysis		
	HR	95% CI	*p* Value	HR	95% CI	*p* Value
Before PSM						
Treatment (LD combined RT vs. Sorafenib)	0.52	0.45–0.59	<0.001	0.43	0.37–0.50	<0.001
Sex (Female vs. Male)	1.11	0.92–1.35	0.275	N.S.		
Age	1.00	1.00–1.01	0.456	N.S.		
ECOG PS (2-3 vs. 0-1)	1.35	1.08–1.70	0.009	N.S.		
Child–Pugh class (B-C vs. A)	1.98	1.71–2.30	<0.001	1.78	1.52–2.07	<0.001
Prior treatment history (Yes vs. No)	1.13	0.98–1.30	0.089	N.S.		
Log(Pretreatment AFP)	1.22	1.16–1.27	<0.001	1.18	1.13–1.24	<0.001
Tumor size	1.04	1.03–1.05	<0.001	1.04	1.03–1.06	<0.001
Disease extent (Bilateral vs. Unilateral)	1.39	1.22–1.58	<0.001	N.S.		
LN status (Involved vs. Not involved)	1.29	1.07–1.55	0.008	N.S.		
PVTT type (III, IV vs. I, II)	1.23	1.08–1.40	0.002	N.S.		
After PSM						
Treatment (LD combined RT vs. Sorafenib)	0.52	0.44–0.61	<0.001	0.46	0.39–0.55	<0.001
Sex (Female vs. Male)	0.98	0.77–1.26	0.887	N.S.		
Age	1.00	0.99–1.01	0.425	N.S.		
ECOG PS (2-3 vs. 0-1)	1.60	1.19–2.14	0.002	N.S.		
Child–Pugh class (B vs. A)	1.88	1.55–2.28	<0.001	1.69	1.38–2.07	<0.001
Prior treatment history (Yes vs. No)	0.91	0.77–1.09	0.308	N.S.		
Log(Pretreatment AFP)	1.27	1.19–1.35	<0.001	1.23	1.15–1.31	<0.001
Tumor size	1.05	1.04–1.07	<0.001	1.05	1.03–1.07	<0.001
Disease extent (Bilateral vs. Unilateral)	1.35	1.14–1.61	<0.001	N.S.		
LN status (Involved vs. Not involved)	1.39	1.08–1.79	0.011	N.S.		
PVTT type (III, IV vs. I, II)	1.28	1.08–1.52	0.004	N.S.		

Abbreviations: PSM, propensity score matching; HR, hazard ratio; CI, confidence interval; LD combined RT, liver-directed combined radiotherapy; ECOG PS, Eastern Cooperative Oncology Group performance status; AFP, alpha-fetoprotein; LN, lymph node; PVTT, portal vein tumor thrombosis; and N.S., not significant.

**Table 3 cancers-15-03164-t003:** Treatment related toxicity in patients treated by sorafenib or liver-directed combined radiotherapy.

	Sorafenib (N = 360)			LD Combined RT (N = 675)		
	Grade 1–2	Grade 3–4	Total	Grade 1–2	Grade 3–4	Total
Acute toxicity (within 3 months)						
Fatigue	21 (5.8%)	0 (0.0%)	21 (5.8%)	13 (1.9%)	2 (0.3%)	15 (2.2%)
Nausea	9 (2.5%)	1 (0.3%)	10 (2.8%)	18 (2.7%)	3 (0.4%)	21 (3.1%)
Vomiting	9 (2.5%)	0 (0.0%)	9 (2.5%)	19 (2.8%)	0 (0.0%)	19 (2.8%)
Anorexia	22 (6.1%)	0 (0.0%)	22 (6.1%)	25 (3.7%)	2 (0.3%)	27 (4.0%)
Fever	4 (1.1%)	2 (0.6%)	6 (1.7%)	14 (2.1%)	0 (0.0%)	14 (2.1%)
Hand–foot syndrome	26 (7.2%)	2 (0.6%)	28 (7.8%)	0 (0.0%)	0 (0.0%)	0 (0.0%)
Skin rash	35 (9.7%)	4 (1.1%)	39 (10.8%)	0 (0.0%)	0 (0.0%)	0 (0.0%)
Leukopenia	0 (0.0%)	0 (0.0%)	0 (0.0%)	5 (0.7%)	0 (0.0%)	5 (0.7%)
Diarrhea	61 (16.9%)	4 (1.1%)	65 (18.1%)	7 (1.0%)	0 (0.0%)	7 (1.0%)
AST/ALT elevation	31 (8.6%)	16 (4.4%)	47 (13.0%)	52 (7.7%)	21 (3.1%)	73 (10.8%)
Bilirubin elevation	27 (7.5%)	15 (4.2%)	42 (11.7%)	38 (5.6%)	26 (3.9%)	64 (9.5%)
Abdominal pain	25 (6.9%)	2 (0.6%)	27 (7.5%)	38 (5.6%)	1 (0.2%)	39 (5.8%)
Late toxicity (after 3 months)						
Fatigue	2 (0.6%)	1 (0.3%)	3 (0.8%)	0 (0.0%)	0 (0.0%)	0 (0.0%)
Hypertension	2 (0.6%)	0 (0.0%)	2 (0.6%)	0 (0.0%)	0 (0.0%)	0 (0.0%)
GI bleeding	13 (3.6%)	10 (2.8%)	23 (6.4%)	0 (0.0%)	5 (0.7%)	5 (0.7%)
Duodenal ulcer	0 (0.0%)	1 (0.3%)	1 (0.3%)	6 (0.9%)	0 (0.0%)	6 (0.9%)

Abbreviations: LD combined RT, liver-directed combined radiotherapy; AST, aspartate aminotransferase; ALT, alanine aminotransferase; and GI, gastrointestinal.

## Data Availability

The data presented in this study are available in this article and the Appendix A.

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
