# Peer review of "Efficacy of Liver-Directed Combined Radiotherapy in Locally Advanced Hepatocellular Carcinoma with Portal Vein Tumor Thrombosis"

_cancers, 2023, doi:10.3390/cancers15123164_

Round 1

Reviewer 1 Report

Interesting study. The authors should explain better how radiotherapy could be used in patients with HCC, given the high risk to determine liver-related adverse events. In fact, it is very difficult to spare the surrounding parenchyma.

The authors should provide a comment on the state of art of loco-regional treatments for hcc, in this regard cite the recent studies PMID: 33339274  and PMID: 34683182 )

English grammar should be improved

Minor grammar improvement is needed

Reviewer 2 Report

In this study, researchers aimed to compare the clinical effectiveness of liver-directed combined radiotherapy with sorafenib, a recommended treatment for locally advanced hepatocellular carcinoma (HCC) with portal vein tumor thrombosis (PVTT). The study involved a multinational patient cohort from ten Asian tertiary hospitals between 2005 and 2014.

A total of 1,035 patients with HCC and PVTT were included, with 675 in the LD-CRT group and 360 in the sorafenib group. To minimize group imbalances, propensity score matching (PSM) was performed, resulting in 305 patients in each group for analysis.

The primary endpoint was overall survival (OS), and secondary endpoints were progression-free survival (PFS) and treatment-related toxicity. After a median follow-up of 22.5 months, the study found that the LD-CRT group had a significantly longer median OS compared to the sorafenib group (10.6 vs. 4.2 months, p < 0.001). Additionally, the LD-CRT group had a higher conversion rate to curative surgery (8.5% vs. 1.0%, p < 0.001) and experienced fewer grade ≥ 3 toxicities (9.2% vs. 16.1%, p < 0.001).

The study concluded that LD-CRT improved survival outcomes and increased the likelihood of conversion to curative surgery in patients with locally advanced HCC presenting PVTT. The authors suggested that active multimodal local treatment involving radiotherapy should be considered for locally advanced HCC with PVTT. However, they also emphasized the need for further prospective studies to validate these findings.

Overall, the study highlights the potential benefits of LD-CRT as a local treatment option for HCC patients with PVTT, offering improved survival outcomes and a higher chance of curative surgery.

1. Figure 1,

this diagram is modified from CONSORT. However, since this is not an RCT.

It's confusing to bring up "CONSORT".

CONSORT is a protocol developed by a group of researchers not only to identify problems arising from conducting RCTs, but also to report, in a full and clear manner, the results yielded by research, thereby facilitating RCTs reading and quality assessment.5 , 6 , 7 It comprises a 25-item checklist focused on scientific article writing (available at www.consort-statement.org). This checklist provides us with standards of how the trial was designed, analyzed and interpreted. Thus, it consists in a useful tool that allows the researcher to conduct a RCT and the clinical orthodontist to critically assess the quality of evidence provided.

Reference

Schulz KF, Altman DG, Moher D. CONSORT Group. CONSORT 2010 Statement: updated guidelines for reporting parallel group randomised trials. BMJ. 2010;340:c332–c332.

Begg C, Cho M, Eastwood S. Improving the quality of reporting of randomized controlled trials. The CONSORT statement. JAMA. 1996;276(8):637–639.

 Moher D, Schulz KF, Altman DG. The CONSORT statement: revised recommendations for improving the quality of reports of parallel-group randomized trials. Ann Intern Med. 2001;134:657–662.

2. The acronym "LD-CRT" liver-directed combined radiotherapy (LD-CRT) may be confused with low-dose conformal radiotherapy or even Chemoradiotherapy.

Please consider modify this to enhance better citation rate to your work.

3. page 5:
PVTT type according to Cheng’s classification?? in which paragraph did you illustrate "Cheng’s classification"

Round 2

Reviewer 1 Report

The revised version is OK. Thank you!